# Peer review of "A Rare Case of TP63-Associated Lymphopenia Revealed by Newborn Screening Using TREC"

_ijms, 2024, doi:10.3390/ijms251910844_

Round 1

Reviewer 1 Report (Previous Reviewer 2)

Comments and Suggestions for Authors

Dear Authors,

Thank you for submitting the revised version of your manuscript. I have carefully reviewed the changes made in response to the previous comments. Overall, you have addressed most of the major concerns raised, and the manuscript has significantly improved. However, there are a few minor points that still need attention:

1. In the expanded immunological data section (Table 1), please consider adding a brief explanation of the significance of the observed changes in T cell subsets over time. This would help readers better understand the clinical implications of these findings.

2. The discussion on the role of TP63 in thymic development (lines 167-183) is informative, but it would be beneficial to include a sentence or two on how this specifically relates to your patient's phenotype.

3. In the genotype-phenotype correlation section (lines 192-205), the newly added Figure 3 is helpful. However, consider adding a brief legend to the figure explaining the domain structure and the significance of the variant locations.

4. While you have explained why functional studies were not conducted, it would be valuable to suggest specific experiments that could be performed in future research to further elucidate the role of this TP63 variant in immune function.

5. The one-year follow-up information (lines 119-121) is a good addition. Consider elaborating slightly on the severity and frequency of the reported infections to give readers a clearer picture of the patient's clinical course.

These minor revisions will further enhance the clarity and impact of your manuscript. I look forward to seeing the final version of your work.

Best regards,

Comments on the Quality of English Language

Minor editing of English language required.

Author Response

The authors extend their gratitude to the Reviewers for the opportunity to submit a revised version of our manuscript for potential publication in the International Journal of Molecular Sciences. We deeply appreciate the time and effort invested in reviewing our work, and we are thankful for the insightful comments and constructive suggestions provided. These have greatly enhanced the quality of our manuscript.

In the revised manuscript, all changes are highlighted using the ‘tracked changes’ feature. We have provided point-by-point responses to each of the reviewers’ comments, which are outlined below and highlighted in green for easy reference.

We sincerely hope that the Reviewers find our responses satisfactory and that the revised manuscript meets the standards for publication. However, we remain open to making further revisions as needed. Thank you once again for your continued interest in our research and for the valuable guidance offered during this process.

Reviewer 1
Thank you for submitting the revised version of your manuscript. I have carefully reviewed the changes made in response to the previous comments. Overall, you have addressed most of the major concerns raised, and the manuscript has significantly improved. However, there are a few minor points that still need attention:

  1. In the expanded immunological data section (Table 1), please consider adding a brief explanation of the significance of the observed changes in T cell subsets over time. This would help readers better understand the clinical implications of these findings.

                A1.1. Thank you for your comment. We have updated the Discussion section accordingly (lines 199-201).

  1. The discussion on the role of TP63 in thymic development (lines 167-183) is informative, but it would be beneficial to include a sentence or two on how this specifically relates to your patient's phenotype.

                A1.2. Thank you for your insightful comment. We have revised the Discussion section (lines 193-198) to better connect existing knowledge on thymus development with the phenotype observed in our patient.

  1. In the genotype-phenotype correlation section (lines 192-205), the newly added Figure 3 is helpful. However, consider adding a brief legend to the figure explaining the domain structure and the significance of the variant locations.

                A1.3. Thank you for your comment. We have revised the Discussion section (lines 209-225) to include the clinical significance of the described variants, along with the results of functional studies where available. Additionally, the domain structure of TP63 is illustrated in Figure 3 for further clarification.

  1. While you have explained why functional studies were not conducted, it would be valuable to suggest specific experiments that could be performed in future research to further elucidate the role of this TP63 variant in immune function.

                A1.4. Thank you for your insightful comment. We have updated the manuscript to include information about the functional consequences of the R343W variant on protein function (line 173). Additionally, we have expanded the discussion to explore potential methods for evaluating the role of TP63 variants in the development and functioning of the immune system (lines 232-237).

  1. The one-year follow-up information (lines 119-121) is a good addition. Consider elaborating slightly on the severity and frequency of the reported infections to give readers a clearer picture of the patient's clinical course.

                A1.5. We have revised the Discussion section to include a detailed examination of the clinical course of the recurrent infections observed in our patient. Based on our analysis, we conclude that these infections are unlikely to be directly associated with immune dysfunction. (lines 193-199).

These minor revisions will further enhance the clarity and impact of your manuscript. I look forward to seeing the final version of your work.

Reviewer 2 Report (Previous Reviewer 1)

Comments and Suggestions for Authors

Thank you for the corrections, I think they made the manuscript more understandable.

I think that to further confirm the effect of the variant, it may be useful to measure gene or protein expression by real time PCR or Westren blot.

Author Response

The authors extend their gratitude to the Reviewers for the opportunity to submit a revised version of our manuscript for potential publication in the International Journal of Molecular Sciences. We deeply appreciate the time and effort invested in reviewing our work, and we are thankful for the insightful comments and constructive suggestions provided. These have greatly enhanced the quality of our manuscript.

In the revised manuscript, all changes are highlighted using the ‘tracked changes’ feature. We have provided point-by-point responses to each of the reviewers’ comments, which are outlined below and highlighted in green for easy reference.

We sincerely hope that the Reviewers find our responses satisfactory and that the revised manuscript meets the standards for publication. However, we remain open to making further revisions as needed. Thank you once again for your continued interest in our research and for the valuable guidance offered during this process.

Reviewer 2
Thank you for the corrections, I think they made the manuscript more understandable.

I think that to further confirm the effect of the variant, it may be useful to measure gene or protein expression by real time PCR or Westren blot.

                A2.1. Thank you for your comment. The R343W variant has already been functionally characterized and shown to significantly impair the transactivation activity of TP63 in a dominant-negative manner. We have elaborated on these findings in the Discussion section (lines 172–174) and referenced the work by Beaudry et al. (2009), cited as reference 24 in our manuscript. In this study, the authors performed a Western blot analysis and a luciferase reporter assay, demonstrating that the mutant protein (designated as R304W in the article) maintained stable expression levels but exhibited a marked reduction in the induction of PERP—a tetraspan membrane protein that plays a crucial role in maintaining the integrity of stratified epithelia (Attardi et al., 2000; Ihrie et al., 2005). This result provides insight into the functional consequences of the R343W variant and its potential impact on the pathogenesis of TP63-related disorders.

Round 2

Reviewer 2 Report (Previous Reviewer 1)

Comments and Suggestions for Authors

Thank you for your reply and your corrections. 

I have no further questions.

This manuscript is a resubmission of an earlier submission. The following is a list of the peer review reports and author responses from that submission.

Round 1

Reviewer 1 Report

Comments and Suggestions for Authors

The authors describe in their case report the substitution of the TP63 gene p.(Arg343Trp) in Ectrodactyly-Ectodermal Dysplasia-Сleft lip/palate syndrome (EEC) syndrome and associated primary immunodeficiency. The detected rare pathogenic variant has already been associated with EEC syndrome in several reported studies. The authors link primary immunodeficiency and the pathogenic variant of the TP63 gene which was found as a new clinical association.

I have the following comments/questions:

1.      In the introduction, the authors do not mention the function of the TP63 gene and the pathogenic mutations of this gene that have been reported so far. A description of this would, in my opinion, be useful to understand the case.

2.      Since the relationship between the TP63 gene and immunodeficiency is poorly understood, the question arises whether the patient has a genetic variant that might affect this, even in interaction with the TP63 gene. Based on the above, I propose to use the WES data to investigate variants in the genes that may be involved.

3.      To support your statement, I think it would be useful to have functional studies that directly link the TP63 gene to immunological changes.

Reviewer 2 Report

Comments and Suggestions for Authors

Dear Authors,

Thank you for submitting your case report on the discovery of a TP63-associated disorder through newborn screening. Your study provides new insights into the potential role of the TP63 gene in immune system development. After careful review, I suggest the following revisions:

Major Revisions:

1. Expansion of Immunological Data:

Please provide more detailed immunological examination results, including complete counts of T cell subsets, B cells, and NK cells. Consider adding a comparison of the patient's immunological data at different age points to better demonstrate the dynamic changes in immune function.

2. TP63 and Immune System Relationship:

Explore in greater depth the specific role of TP63 in thymic development and T cell generation. Consider adding a brief literature review summarizing the currently known functions of TP63 in the immune system.

3. Genotype-Phenotype Correlation:

Discuss in detail the relationship between the p.(Arg343Trp) mutation and the patient's phenotype, particularly focusing on the connection with immune system abnormalities. Compare this mutation with other reported TP63 mutations leading to immune phenotypes.

4. Functional Studies:

If possible, consider conducting in vitro functional studies to validate the effect of this mutation on TP63 protein function, especially in immune-related cells.

Minor Revisions:

1. Methodological Details:

Please provide more technical details about the TREC/KREC detection and whole-exome sequencing, including the instruments and analysis software used.

2. Clinical Follow-up:

Add information on long-term follow-up of the patient, including growth and development, infection history, and other relevant clinical manifestations.

Overall, your study provides valuable clues for understanding the role of TP63 in immune system development. With these revisions, I believe your article will be more comprehensive and impactful. I look forward to receiving your revised manuscript.

Best wishes for your research!

Comments on the Quality of English Language

Minor editing of English language required.